# Bioanalytical System for Determining the Phenol Index Based on *Pseudomonas putida* BS394(pBS216) Bacteria Immobilized in a Redox-Active Biocompatible Composite Polymer “Bovine Serum Albumin–Ferrocene–Carbon Nanotubes”

**DOI:** 10.3390/polym14245366

**Published:** 2022-12-08

**Authors:** Roman N. Perchikov, Daria V. Provotorova, Anna S. Kharkova, Vyacheslav A. Arlyapov, Anastasia S. Medvedeva, Andrey V. Machulin, Andrey E. Filonov, Anatoly N. Reshetilov

**Affiliations:** 1Federal State Budgetary Educational Institution of Higher Education, Tula State University, 300012 Tula, Russia; 2Federal State Budgetary Institution of Science, N.D. Zelinsky Institute of Organic Chemistry, 119991 Moscow, Russia; 3Federal Research Center «Pushchino Scientific Center for Biological Research of the Russian Academy of Sciences», G.K. Skryabin Institute of Biochemistry and Physiology of Microorganisms, Russian Academy of Sciences, 142290 Pushchino, Russia

**Keywords:** phenol index, biosensor, redox-active polymer, bacteria, carbon nanotubes, bovine serum albumin-ferrocene-carbon nanotubes

## Abstract

The possibility of using the microorganisms *Pseudomonas* sp. 7p-81, *Pseudomonas putida* BS394(pBS216), *Rhodococcus erythropolis* s67, *Rhodococcus pyridinivorans* 5Ap, *Rhodococcus erythropolis* X5, *Rhodococcus pyridinivorans* F5 and *Pseudomonas veronii* DSM 11331^T^ as the basis of a biosensor for the phenol index to assess water environments was studied. The adaptation of microorganisms to phenol during growth was carried out to increase the selectivity of the analytical system. The most promising microorganisms for biosensor formation were the bacteria *P. putida* BS394(pBS216). Cells were immobilized in redox-active polymers based on bovine serum albumin modified by ferrocenecarboxaldehyde and based on a composite with a carbon nanotube to increase sensitivity. The rate constants of the interaction of the redox-active polymer and the composite based on it with the biomaterial were 193.8 and 502.8 dm^3^/(g·s) respectively. For the biosensor created using hydrogel bovine serum albumin-ferrocene-carbon nanotubes, the lower limit of the determined phenol concentrations was 1 × 10^−3^ mg/dm^3^, the sensitivity coefficient was (5.8 ± 0.2)∙10^−3^ μA·dm^3^/mg, Michaelis constant K_M_ = 230 mg/dm^3^, the maximum rate of the enzymatic reaction R_max_ = 217 µA and the long-term stability of the bioanalyzer was 11 days. As a result of approbation, it was found that the urban water phenol content differed insignificantly, measured by creating a biosensor and using the standard photometric method.

## 1. Introduction

One of the dangerous and most common environmental pollutants is phenol and its derivatives, including volatile ones, which have a toxic effect on living organisms. These substances are formed as a result of the activities of enterprises associated with the processing of petroleum products and coal, with the production of dyes, pharmaceuticals, and plasticizers [1,2]. In surface waters, dissolved phenols exist both in free form and in the form of phenolate ions [3]. Phenols can cause diseases of the nervous system and kidneys; phenol and its metabolites can damage DNA and enzymes, as well as have a carcinogenic, hepatotoxic effect [4,5,6,7], which can contribute to the rapid death of aquatic organisms.

The content of phenols in water is determined by standard methods such as photometry [8] and fluorimetry [9]. The trend towards cheaper, simpler, and more automated methods of analysis used in modern chemical and biochemical laboratories has led to the creation of biosensor-based instruments for the determination of phenol. The following enzymes are used in the development of biosensors for the determination of phenolic compounds: tyrosinase [10,11,12], laccase [13,14], and peroxidase [15,16]. Such sensors are quite selective [17]. The main disadvantage of enzyme biosensors for phenol detection is the low stability of the analytical signal due to the inactivation of the biomaterial by reaction metabolites or other pollutants present in the sample; in addition, enzymes are often characterized by a narrow temperature optimum, which has a significant effect on the generated analytical signal.

Due to the disadvantages of enzymes, a large number of microbial analogues are being developed. The following microorganisms for the detection of phenolic compounds have been noted: *Moraxella* [17], *Arthrobacter* [18], *Pseudomonas* [19,20], *Trichosporon* [21], *Rhodococcus* [22], *Lactobacillus* [23] and *Escherichia* [24]. Microbial biosensors, as a rule, are inferior to enzymatic biosensors in sensitivity. Thus, in the article [17], the microorganisms *Moraxella * sp. Are cultivated in a mineral medium containing yeast extract and para-nitrophenol. The presented biosensor for the determination of p-nitrophenol, based on a graphite-paste electrode modified by *Moraxella * sp. microorganisms, is superior to the biosensor based on tyrosinase from the article [11]. The limits of detection are 20 nM and 130 nM, respectively. The upper limit of the linear portion of the calibration dependence for the microbial biosensor is 20 μM. The use of a small amount of biomass in the manufacture of the electrode and high stability make the use of microbial sensors preferable over enzymatic ones [25]. The biosensor presented in paper [17] was developed on the basis of an oxygen electrode, which can lead to an overestimation of the result due to the oxidation of other substrates; in this case, the lower limit of the determined concentrations will be overestimated. 

However, the main problem of microbial biosensors is their low selectivity towards phenolic compounds, which appears due to the wide substrate specificity of the biomaterial. To solve this problem, researchers are adapting microorganisms when growing a culture used in biosensor analysis. This approach makes it possible to use natural strains of microorganisms without carrying out their genetic modification, while the selectivity of determining the phenol index increases. For example, the authors of paper [26] isolated bacteria of the genus *Pseudomonas* from oil-contaminated soil and adapted microorganisms for the conductometric detection of phenol in order to increase the selectivity of the test system. On the surface of a gold electrode modified with carbon nanotubes, the cells were immobilized in cross-linked bovine serum albumin to increase the lifetime of the biomaterial. The range of determined phenol concentrations was 1–300 mg/L (10–3187 μM) using the microbial biosensor developed by the authors.

The scientific issues of the work are increasing biosensor selectivity by adapting microorganisms to the phenol, improving the biosensor sensitivity and the long-term stability of the biomaterial in receptor elements by using a redox-active polymer and carbon nanotubes composite. An important characteristic of the developed receptor elements is long-term stability. The most effective approach to improving this parameter is an application of conductive or redox-active polymers [27,28]. The redox-active polymer performs a dual function. First, the polymer retains the biomaterial by forming a network structure [29] and pore size, which contributes to the immobilization of microorganisms. Second, due to the presence of a covalently bound mediator in the polymer structure, the matrix provides electron transport from the active sites of microorganism enzymes to the electrode [30,31]. Another problem of phenol biosensor development is low sensitivity. The lower limit of the determined concentration is not suitable for maximum permissible phenol analysis. In this work, a method of increasing sensitivity was studied by using a redox-active polymer and carbon nanotubes composite. Carbon nanotubes increase the effective surface area of the electrode; therefore, the conductivity of the prepared electrode greatly improves and increases the recorded analytical signal and the measurement sensitivity.

The aim of this study was the development of a selective and highly sensitive receptor system based on microorganisms immobilized in a redox-active polymer for rapid detection of the phenol index.

## 2. Materials and Methods

### 2.1. Strains of Microorganisms

Bacteria *Pseudomonas * sp. 7p-81 and *Pseudomonas putida* BS394(pBS216), *Rhodococcus erythropolis* s67, *Rhodococcus pyridinivorans* 5Ap, *Rhodococcus erythropolis* X5 and *Rhodococcus pyridinivorans* F5 were provided by the Laboratory of Biology of Plasmids of the Institute of Biochemistry and Physiology of Microorganisms of the Russian Academy of Sciences, Federal Research Center Pushchino Center for Biological Research for use in this study. A strain of *Pseudomonas veronii* DSM 11331^T^ was previously isolated from activated sludge [32].

### 2.2. Cultivation of Microorganisms for Biosensor Analysis

The following substances were used for cultivation and adaptation of microorganisms to phenol: ammonium sulfate ((NH_4_)_2_SO_4_, 99%, Diaem), magnesium sulfate (MgSO_4_7H_2_O, 99%, Diaem), potassium dihydrogen phosphate (KH_2_PO_4_, 99%, Diaem), potassium hydrogen phosphate (K_2_HPO_4_, 99%, Diaem), calcium chloride (CaCl_2_2H_2_O, 99%, Diaem), manganese sulfate (MnSO_4_5H_2_O, 99%, Diaem), phenol (C_6_H_5_OH, 99%, Diaem) and glucose (C_6_H_12_O_6_H_2_O, 99%, Diaem).

The cultivation of microorganisms was carried out at different stages of adaptation on a mineral medium with the following composition (%):ammonium sulfate ((NH_4_)_2_SO_4_, 99%, Diaem)—0.03, magnesium sulfate (MgSO_4_7H_2_O, 99%, Diaem)—0.01, potassium hydrogen phosphate (K_2_HPO_4_, 99%, Diaem)—0.26, potassium dihydrogen phosphate (KH_2_PO_4_, 99%, Diaem)—0.14, calcium chloride (CaCl_2_2H_2_O, 99%, Diaem)—0.001 and manganese sulfate (MnSO_4_5H_2_O, 99%, Diaem)—0.002, with glucose (C_6_H_12_O_6_H_2_O, 99%, Diaem) 240 mg/dm^3^ and with different content of phenol in test tubes with a capacity of 50 cm^3^ for 24 h at a temperature of 28 °C. The resulting biomass was separated by centrifugation for 10 min at 10,000 rpm and washed with potassium sodium phosphate buffer pH 6.8. Then, fresh portions of the buffer were added into microtubes and precipitated in an Eppendorf centrifuge for 10 min at 10,000 rpm. The biomass was stored in microtubes at 20 °C.

### 2.3. Adaptation of Microorganisms to Phenol

An adaptation of microorganisms to phenol was carried out in a mineral medium in test tubes with a capacity of 50 cm^3^ gradually. Next, 1 cm^3^ of the inoculum was transplanted into 25 cm^3^ of a mineral medium containing 10 mg/cm^3^ of phenol and 240 mg/dm^3^ of glucose. The phenol concentration was increased and glucose concentration was reduced by each reinoculation. The final concentration of phenol was 250 mg/dm^3^. The studies were carried out at different stages of adaptation of microorganisms to phenol at the following concentrations of phenol in the growth medium: for *P. veronii* DSM 11331^T^ and for *R. pyridinivorans* 5Ap—50 mg/dm^3^, 150 mg/dm^3^ and 250 mg/dm^3^ and for *P. putida* BS394(BS216)—30 mg/dm^3^, 130 mg/dm^3^ and 250 mg/dm^3^.

### 2.4. Synthesis of a Redox-Active Polymer Based on Bovine Serum Albumin (BSA) and a Neutral Red Mediator (BSA-NR)

The redox-active polymer synthesis procedure was described in [30]. First, 3.5 mg of bovine serum albumin (BSA) was solved in 50 μL of phosphate buffer (pH = 6.8), then 5 μL of 0.6 M neutral red was added. At last, 7.5 µL of glutaraldehyde was added and the solution was shaken for 30 s. Then, 10 μL produced polymer was dropped onto the cleaned surface of the carbon-paste electrode and left to dry.

#### Synthesis of a Redox-Active Polymer Based on BSA and the Mediator Ferrocenecarboxaldehyde (BSA-FC)

The redox-active polymer synthesis procedure was described in [30]. Two solutions were mixed: the first was 0.5 g of BSA in 5 mL phosphate buffer (pH = 6.8) and the second was 0.05 g ferrocenecarboxaldehyde in 5 mL of acetone. Then, 5% K_2_CO_3_ was added while pH = 9.3 solution was not achieved. Interaction between BSA and ferrocenecarboxaldehyde was carried out for one hour at room temperature, then 10 mg of NaBH_4_ was added to the mixture. Then, the solution was kept at room temperature for 6 h. The resulting mixture pH was adjusted to 6.5 for borohydride decomposition. Then, the mixture pH was adjusted to 8.5 by adding 0.1 M NaOH solution to precipitate the polymer. The solution with the polymer was centrifuged at 3000 rpm for 20 min. The resulting supernatant was dialyzed with phosphate buffer (pH = 6.8) for several days at 4 °C to separate unreacted ferrocenecarboxaldehyde. To obtain BSA-FC electrode, 3.5 mg the synthesized polymer was solved in 50 µL of phosphate buffer (pH = 6.8) and 7.5 µL of glutaraldehyde was added for 30 s. Then, 10 µL of the resulting mixture was added to the electrode.

### 2.5. Formation of the Working Electrode with Immobilized Microorganisms

Graphite powder (Fluka, Dresden, Germany), mineral oil (Helicon, Moscow, Russia), acetone, dialysis membrane with a transmission limit of 14 kDa (Roth, Germany), ferrocene (C_10_H_10_Fe, 98%, Sigma-Aldrich, Darmstadt, Germany), ferrocenecarboxaldehyde (C_10_H_9_FeCHO, 98%, Sigma-Aldrich, Germany), neutral red (C_15_H_17_ClN_4_, >97%, Diaem, Moscow, Russia), thionine (C_12_H_10_N_3_S, 98%, Diaem, Russia), methylene blue (C_16_H_18_ClN_3_S, 98%, Diaem, Russia) and potassium hexacyanoferrate (III) (K_3_[Fe(CN)_6_], 99%, Diaem, Russia) were used to form a working graphite-paste electrode. Single-walled carbon nanotubes (SWCNT) (NPF OOO Uglerod ChG, Russia, Chernogolovka; water-based dispersion 2.5% SWCNT) were used to form an electrode modified with the redox-active polymer BSA-FC/CNT.

To form a working electrode based on neutral red (NR), thionine, methylene blue and potassium hexacyanoferrate (III), a plastic tube was filled with graphite paste and 100 mg of graphite powder and 40 μL of mineral oil were mixed. Then, 10 μL of microorganism suspension (0.33 g/cm^3^) was applied to the electrode surface, dried at room temperature, and a dialysis membrane was fixed on the electrode with a plastic ring. NR, thionine, methylene blue and potassium hexacyanoferrate (III) were added to phosphate buffer (pH = 6.8) in which measurements were carried out.

To form a working electrode based on ferrocene (FC) or ferrocenecarboxaldehyde, a plastic tube was filled with a modified graphite paste to obtain 90 mg of graphite powder, then 10 mg of a mediator, 40 μL of mineral oil, and 500 μL of acetone were mixed. Next, 10 μL of a suspension of microorganisms (0.33 g/cm^3^) was applied to the electrode surface, dried at room temperature, and a dialysis membrane was fixed on the electrode with a plastic ring.

When forming an electrode based on redox polymers BSA-NR, BSA-FC, BSA-FC/CNT, a plastic tube was filled with graphite paste to obtain a graphite paste and 100 mg of graphite powder and 40 μL of mineral oil were mixed. Following that, 10 µL of the redox active polymer BSA-FC or BSA-NR (0.07 g/cm^3^) was applied to the electrode surface, then 10 µL of a suspension of microorganisms (0.33 g/cm^3^) was dried and fixed on the electrode with a dialysis membrane using a plastic ring. When modifying the electrode with the redox-active BSA-FC/CNT polymer, 10 μL of a CNT suspension (specific density 2.5 μg/mm^2^) was preliminarily applied, and then the BSA-FC matrix was applied.

### 2.6. Registration of Current-Voltage Dependences

An Ecotest-VA voltammetric analyzer (Ekoniks-Expert, Moscow, Russia) was used to record a voltammogram. A three-electrode cell was applied. A graphite-paste electrode with immobilized microorganisms was used as a working electrode, and a platinum electrode was used as a counter electrode. A saturated silver chloride electrode (Ag/AgCl) was used as a reference electrode. Cyclic voltammograms were recorded at a scan rate of 20–100 mV/s in phosphate buffer (pH = 6.8).

### 2.7. Biosensor Measurements

A two-electrode cell was connected to IPC-micro galvanopotentiostat (Volta, Moscow, Russia). A graphite-paste electrode with immobilized microorganisms was used as a working electrode. A saturated silver chloride electrode (Ag/AgCl) was used as a reference one. The measurements were carried out at a temperature of 20 °C in phosphate buffer (pH = 6.8). The current change after adding the analyte was taken as the biosensor response.

### 2.8. Scanning Electron Microscopy (SEM)

A JEE-4X vacuum deposition unit (JEOL, Japan) was used to deposit a thin layer of a plati-num–carbon mixture onto the matrix samples. A JSM-6510 LV scanning electron microscope (JEOL, Tokyo, Japan) was applied to produce scanning electron microscopy images of the samples in a high vacuum mode during registration of secondary electrons.

### 2.9. Determination of Phenol by Photometric Method

For the determination of phenol by the standard method, copper (II) sulfate (CuSO_4_5H_2_O, 98%, Diaem), sulfuric acid (H_2_SO_4_, 99%, Diaem), sodium hydroxide (NaOH, 99%, Diaem), 4-aminoantipyrine (C₁₁H₁_3_N_3_O, 98%, Diaem), potassium hexacyanoferrate (III) (K_3_[Fe(CN)_6_], ≥99%, Diaem), chloroform (CHCl_3_, ≥98%, Diaem) and phenol (C_6_H_5_OH, ≥99%, Diaem) were used. As a reference method of analysis for the determination of volatile phenols, we used the photometric method after distillation with water vapor in accordance with the procedure [8]. Active chlorine was determined by the iodometric method [33]. The measurements were carried out using a PE 5400-UF spectrophotometer (EKROSKHIM, Saint-Petersburg, Russia).

## 3. Results

### 3.1. Selectivity of Microorganisms in the Adaptation Process

It is necessary that the biomaterial be as selective as possible for phenol in order to determine the phenol index using microbial biosensors. Seven strains of microorganisms were used as a biomaterial for the biosensor formation. They are *Pseudomonas * sp. 7p-81, *Pseudomonas putida* BS394(pBS216), *Pseudomonas veronii* DSM 11331^T^, *Rhodococcus erythropolis* s67, *Rhodococcus pyridinivorans* 5Ap, *Rhodococcus erythropolis* X5 and *Rhodococcus pyridinivorans* F5. Bacteria of the genus *Rhodococcus* are able to efficiently oxidize aromatic compounds [34], and bacteria of the genus *Pseudomonas* are used in the development of biosensors [35], as they are natural hosts of many catabolic plasmids—NAH, TOL, CAM, OCT, SAL and pBS216; the latter provides the process of biodegradation of naphthalene, which determines the oxidation of this compound to metabolites of the Krebs cycle. *Pseudomonas* are good recipients of catabolic plasmids and they are used for bioremediation of oil pollution [36].

The use of ferrocene as a mediator is due to the fact that the occurrence of electrochemical reactions involving ferrocene does not depend on the pH of the medium. Poor solubility of ferrocene in water makes it possible to produce stable carbon paste modification [37] and provides no reagent analysis [35,38]. All studied microorganisms are able to interact with an artificial electron acceptor (ferrocene), which is confirmed by the presence of biosensor responses at an operating potential of 250 mV. This potential is most often used with ferrocene-modified graphite-paste electrodes with immobilized cells of microorganisms.

The selectivity of bacteria was evaluated. Strains of *Pseudomonas * sp. 7p-81, *Rhodococcus erythropolis* s67, *Rhodococcus erythropolis* X5 and *Rhodococcus pyridinivorans* F5 were not promising for the analysis of phenol, as they demonstrated a wide substrate specificity, and phenol oxidation was not registered (Figure 1).

Bacteria *P. veronii* DSM 11331^T^, *P. putida* BS394(pBS216) and *R. pyridinivorans* 5Ap oxidize phenol under electrocatalytic conditions, but receptor systems based on them have low selectivity for assessing the phenol index. Therefore, the influence of adaptation due to the growth of microorganisms on the selectivity of the analytical system was investigated (Figure 2a–c).

Non-adapted *Rhodococcus pyridinivorans* 5Ap oxidize phenol at low concentrations (0.4 mM). In the process of adaptation of microorganisms, substrate specificity narrows; the response to sugars, alcohols and amino acids decrease, and a response to 1 M phenol and its derivatives also appears (Figure 2a).

Changes in the substrate specificity of microorganisms *P. veronii* DSM 11331^T^ occur as follows: during adaptation at the initial stages, an increase in responses to sugars was revealed, but with an increase in the concentration of phenol in the growth medium, they decrease. This is probably due to the fact that during the adaptation, microbes produce more oxidoreductase enzymes [19,39,40] (Figure 2b). With an increase in the concentration of phenol in the culture medium from 0 to 250 mg/dm^3^, a decrease in responses to other substrates is observed: alcohols, amino acids, organic acids and their salts. In general, when adapting, *P. veronii* DSM 11331^T^ managed to reduce the amount of oxidizable substrates from 25 to 11.

Bacteria *P. putida* BS394(pBS216) reduced the amount of oxidizable substrates faster during adaptation (from 25 to 17 substrates), and at the end of adaptation, the ability to oxidize phenol and p-nitrophenol increased in contrast to *P. veronii* DSM 11331^T^ (Figure 2c). In terms of selectivity, the adapted bacteria *P. putida* BS394(pBS216) are more promising for further use in the development of a biosensor for the determination of phenol.

### 3.2. Metrological Characteristics of Receptor Systems Based on the Microorganisms Pseudomonas veronii DSM 11331^T^, Pseudomonas putida BS394(pBS216) and Rhodococcus pyridinivorans 5Ap

Calibration dependences of the biosensor response on the phenol concentration were plotted at different stages of the adaptation of microorganisms *P. veronii* DSM 11331^T^ (Figure 3a), *P. putida* BS394(pBS216) (Figure 3b) and *R. pyridinivorans* 5Ap (Figure 3c) to determine the metrological characteristics of receptor elements.

The dependence of the biosensor response on the substrate concentration was approximated by the Michaelis–Menten equation:(1)$$R=\frac{R_{max}\left[S\right]}{K_{M}+\left[S\right]},$$
where *R* is the rate of the enzymatic reaction; *R_max_* is the maximum rate of the enzymatic reaction; [*S*] is the substrate concentration; *K_M_* is the apparent Michaelis constant.

The main metrological and kinetic characteristics of receptor elements at different stages of adaptation are presented in Table 1.

Similar tendencies are observed in changing the characteristics of biosensors in the process of adaptation of microorganisms: at first, there is an increase in the value of the apparent Michaelis constant, which indicates that the affinity of cells to phenol decreases. Probably, at the beginning, phenol has a toxic effect on microorganisms, but at the end of adaptation, the affinity for phenol in both strains increases. The Michaelis constant is lower than that of non-adapted microorganisms, indicating that the cells have adapted to phenol oxidation. In the process of adaptation of *P. putida* BS394(pBS216) and *R. pyridinivorans* 5Ap, a decrease in the lower limit of the determined concentrations is observed. Microorganisms *P. putida* BS394(pBS216) are able to oxidize lower concentrations of phenol, so they were chosen for further study.

### 3.3. Selection of Redox-Active Particles and Redox-Active Polymers for the Formation of a Bioreceptor System

General regularities in the efficiency of charge transfer using redox-active particles from adapted microorganisms to the working surface of the indicator electrode were revealed. For this, a comparative evaluation of the efficiency of bioelectrocatalytic phenol oxidation by *Pseudomonas putida* BS394(pBS216) microorganisms in the presence of various redox compounds was carried out. Neutral red, thionine, methylene blue, potassium hexacyanoferrate (III), ferrocene and ferrocenecarboxaldehyde were used as redox-active compounds.

When recording a cyclic voltammogram in the presence of phenol, the anode current increases (Figure 4) due to the electrocatalytic oxidation of the substrate by bacteria, which proceeds according to the following scheme:S + E_ox_ (*P. putida* BS394(pBS216)) → P + E_red_ (*P. putida* BS394(pBS216))(2)
(3)$$E_{\mathrm{red}}\;\left(P.\;putida\;\mathrm{BS}394\left(\mathrm{pBS}216\right)\right)+\;M_{\mathrm{ox}}\;\overset{k_{in}}{\Rightarrow }M_{\mathrm{red}}\;+\;E_{\mathrm{ox}}\;\left(P.\;putida\;\mathrm{BS}394\left(\mathrm{pBS}216\right)\right)$$
M_red_ → M_ox_ + nē(4)
where S is the substrate (phenol); P is product; E_ox_ (*P. putida* BS394(pBS216)) and E_red_ (*P. putida* BS394(pBS216)) are oxidized and reduced forms of the bacterial enzyme *P. putida* BS394(pBS216); M_red_ and M_ox_ are the reduced and oxidized form of the electron transport mediator, nē is the number of electrons transferred to the electrode; *k_in_* is the rate constant of the interaction between the mediator and the biomaterial.

In the case of high-rate constant kin in Equation (2), the mediator successfully competes with dissolved oxygen and biosensor analysis does not require deaeration. For rate constant kin measurement, the Nicholson and Shine approach [41] was used. According to the method, excess amount of substrate was used; in this case reaction (1) is not limiting, and the diffusion control of the limiting current was checked. In this case reaction, (3) is not rate-determining. To determine the rate constants, the dependences of the ratio of the limiting anode currents in the presence and in the absence of the substrate (*I_k_*/*I_d_*) on the value 1/*ν*^0.5^ were obtained; *k_in_* was found using Equation (5) by the tangent of the slope of the linear regression according to the graph shown in the (Figure 5).
(5)$$\frac{I_{k}}{I_{d}}=\sqrt{\frac{k_{in}\left[E\right]RT}{nF\nu }},$$
where *I_k_* is the limiting current after substrate addition; *I_d_* is the limiting current before substrate addition (A); *k_in_* is the rate constant of the reaction (2) (dm^3^/mg × s); *R* is the universal gas constant, J/mol × K; *T* is temperature, deg. Kelvin K; [*E*] is cell suspension titer (mg/dm^3^); *ν*—scan rate (V/s); *n* is the number of electrons; *F* is Faraday’s constant (C/mol).

The obtained values of the interaction rate constants of mediators with bacteria *P. putida* BS394(pBS216) are presented in Table 2. In terms of the rate constants of interaction of mediators of electron transport, the best acceptor properties were demonstrated by the mediator neutral red and ferrocene; therefore, these mediators are most promising for the synthesis of a redox polymer, which makes it possible to increase the sensitivity to phenol. The heterogeneous rate constant of electron transfer to the electrode for ferrocene and NR differs by an order of magnitude. Since a higher value of the constant is observed for ferrocene, this mediator is more promising due to the shorter response time of the biosensor.

Bovine serum albumin was chosen as the basis for the redox polymer, due to its high biocompatibility and non-toxicity. The process of synthesis of polymer matrices with selected redox particles was carried out by the mechanism of nucleophilic addition, in which glutaraldehyde played the role of a crosslinking agent (Figure 6).

The structures of the resulting mediator-modified matrices were previously studied by IR spectroscopy and atomic absorption spectroscopy [42]. Changes in the electrochemical properties of a graphite-paste electrode modified with a conductive gel and carbon nanotubes were studied by cyclic voltammetry.

There is a change in the electrochemical properties in the process of modification of the electrode (Figure 7). The addition of CNTs to the system sharply increases the efficiency of electron transfer; as a result, the anodic and cathodic peaks increase. After cell immobilization, the anodic peak reduces, since some of the redox particles also interact with the biomaterial. After adding phenol, the anodic peak increases, which indicates the efficiency of bioelectrocatalysis in the systems under study.

The structure of the studied conducting hydrogels was studied by scanning electron microscopy. The resulting images are shown in Figure 8.

The porous structure of redox-active polymers is shown in Figure 8. The redox-active polymers have pores 10–30 micron in size. This structure allows *P. putida* bacteria to be immobilized into this polymer. Carbon nanotubes application significantly increases the conductive area. Their interposition in the redox-active polymer is given in Figure 8c,d. After immobilization, cells are inside the redox-polymer structure (Figure 8e,f). The possible electron transfer mechanism providing biosensor response generation is shown in Figure 9.

The rate constants of the interaction of neutral red and ferrocene with microorganisms were found (Table 2), from which it follows that the process of electron transfer is most effective when using the redox-active polymer BSA-ferrocene. The introduction of CNTs into the system increases the rate constant of the interaction of bacteria with the redox particles of the polymer. Thus, a nanocomposite redox-active polymer BSA-ferrocene-CNT was chosen to form the biosensor receptor system for phenol determination, which has high electrochemical characteristics.

### 3.4. Characterization of the Receptor Element Based on Pseudomonas putida BS394(pBS216) Microorganism Modified with Redox-Active Polymer BSA-Ferrocene and CNT

The calibration curve of the biosensor responses on the phenol concentration is shown in Figure 10 for the receptor element based on microorganisms *P. putida* BS394(pBS216), which was adapted to phenol and immobilized with a redox-active polymer BSA-ferrocene, together with single-walled CNTs.

Figure 11 shows the substrate specificity for phenol-adapted microorganisms *P. putida* BS394(pBS216) immobilized on a graphite-paste electrode using a redox-active polymer BSA-ferrocene and using single-walled CNT.

After adding CNTs, the change in the selectivity of the receptor element based on microorganisms *P. putida* BS394(pBS216) practically does not occur (Figure 11). The stability and duration of operation of a biosensor based on microorganisms *P*. *putida* BS394(pBS216) immobilized in a gel based on ferrocenecarbaldehyde-modified BSA using single-walled CNTs was determined for reusability limitation estimation. Phenol can be rather aggressive as analyte for bioreceptor. Reversibly, bioanalyte-biorecognition complex kinetics was estimated by relative standard deviation for 15 consecutive measurements. This parameter was 15% (operational stability), which is higher than for other microbial biosensors [30,32], but biosensor response was stable (signals are not decreasing), therefore it can be concluded that the interaction between the bioreceptor and phenol is reversible and the developed system is reusable. Long-term stability was measured over 15 days for different electrodes produced at different times, but to obtain the desired results, when the relative standard deviation of 15 consecutive measurements is not more than 15%, the sensor should be used for only 11 days (long-term stability of the biosensor), after this period the sensor needs to be replaced. A comparison of the present study results with other similar findings is given in Table 3.

For microorganisms immobilized in a gel based on BSA modified with ferrocenecarboxaldehyde, the Michaelis constant is higher by two orders of magnitude, in contrast to bacteria immobilized without using a polymer. The lower limit of the determined concentrations is less by three orders of magnitude. This may be due to the fact that ferrocene integrates directly at a closer distance into the polyenzymatic system of bacteria and accelerates the rate of electron transfer resulting from various enzymatic reactions.

The developed biosensor is inferior to the biosensor-based duroquinone mediator system and bacteria *Pseudomonas * sp. 74-III due to its limited detection of phenol (it is lower by one order), but long-term stability for researching biosensors is higher at 5 days. An increase in long-term stability is achieved due to the redox-active polymer application, which is biocompatible for living systems; on the other hand, covalent bonding of a polymer with a mediator prevents ferrocene washing out during sensor exploitation. The limit of detection for other literature analogues is higher, in particular for the biosensor [44], which is related to using adapted bacteria *P. putida* BS394(pBS216) against a non-adapted consortium [44]. The lower limit of the determined concentrations of the developed biosensor makes it possible to determine the amount of phenols within the maximum allowable concentrations (MAC).

### 3.5. Measurement of the Phenolic Index of Real Samples by the Standard Method

An analysis of the river water samples and model samples, prepared by the addition of phenol concentrations of 0.1, 5 and 24 mg/dm^3^ to surface waters, was carried out. The results are given in the Table 4.

After analyzing a sample of river water, it was not possible to determine the low content of volatile phenols, since the negative influence of environmental factors (the presence of substances that are inhibitors) is possible. The results of the analysis of model samples using biosensor and standard methods differ insignificantly (statistical processing of the results was carried out using a modified Student’s test). Therefore, the developed biosensor can be used as an alternative to the standard method for the analysis of high-concentration phenols.

## 4. Conclusions

As a result of the study of the possibility of using seven strains of bacteria: *Rhodococcus erythropolis* s67, *Rhodococcus pyridinivorans* 5Ap, *Rhodococcus erythropolis* X5, *Rhodococcus pyridinivorans* F5, *Pseudomonas veronii* DSM 11331^T^ and *Pseudomonas * sp. 7p-81, for the evaluation of the phenol index, the most promising bacterial strain *Pseudomonas putida* BS394(pBS216) was identified together with the mediator ferrocene. Changes in the characteristics of the electrode during the adaptation of microorganisms were studied. It was found that adaptation leads to an increase in the selectivity and sensitivity of the bioelectrode to the analysis of the phenol index. The lower limit of determined phenol concentrations for *Pseudomonas putida* BS394(pBS216) was decreased in the process of adaptation from 41.5 mg/dm^3^ to 0.5 mg/dm^3^.

The most effective acceptors of electrons from biomaterial are neutral red and ferrocene. The rate constants of the interaction of these mediators with *Pseudomonas putida* BS394(pBS216) are 182.1 and 97.9 dm^3^/(g∙s), respectively. The synthesized polymers based on BSA modified with ferrocecarboxaldehyde have a porous structure; the size of the pore is from 10 to 30 µm, which allows immobilizing microorganisms into the matrix. Redox-active polymers and nanocomposites are more effective acceptors of electrons from *Pseudomonas putida* BS394(pBS216): the rate constants of interaction between the BSA-FC and the BSA-FC-CNT polymers are 193.8 and 502.8 dm^3^/(g∙s), correspondingly.

The high sensitivity of the biosensor (the lower limit of the determined concentrations is 0.001 mg/L) indicates the possibility of using the developed system as an analogue of the standard method for the determination of phenols in various samples, both natural and wastewaters. The analysis of water samples with a low phenol content showed the need to increase the stability of the bioanalytical system; when analyzing high phenol values, the results of the biosensor determination did not differ significantly from the results of the standard method. The developed biosensor can be used for 11 days, the relative standard deviation of 15 consecutive measurements was 15%.

## Figures and Tables

**Figure 1 polymers-14-05366-f001:**
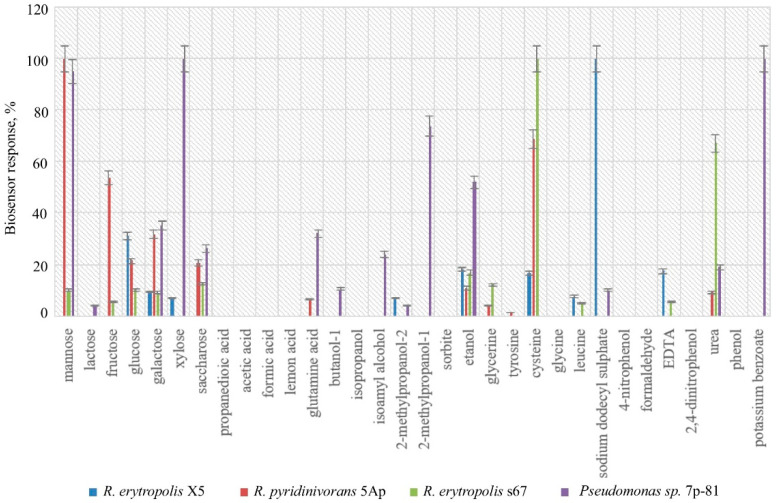
Substrate specificity of some strains of microorganisms of the genera *Rhodococcus* and *Pseudomonas* that are not adapted to phenol.

**Figure 2 polymers-14-05366-f002:**
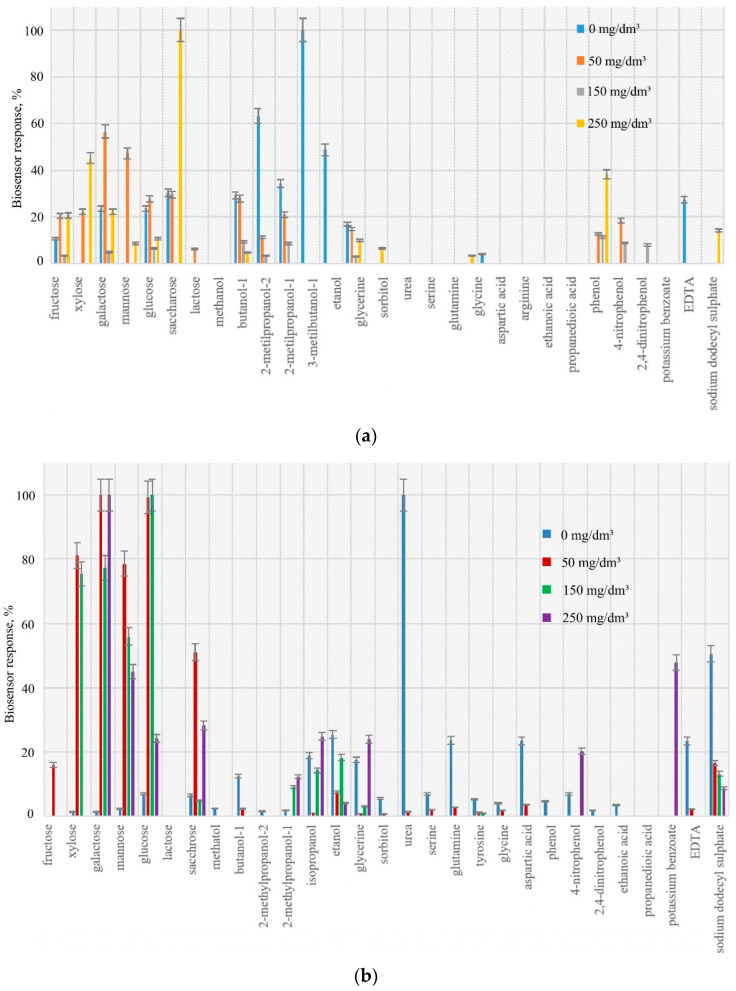
Substrate specificities of microorganisms at different stages of adaptation: (**a**) *R. pyridinivorans* 5Ap; (**b**) *P. veronii* DSM 11331^T^; (**c**) *P. putida* BS394(pBS216).

**Figure 3 polymers-14-05366-f003:**
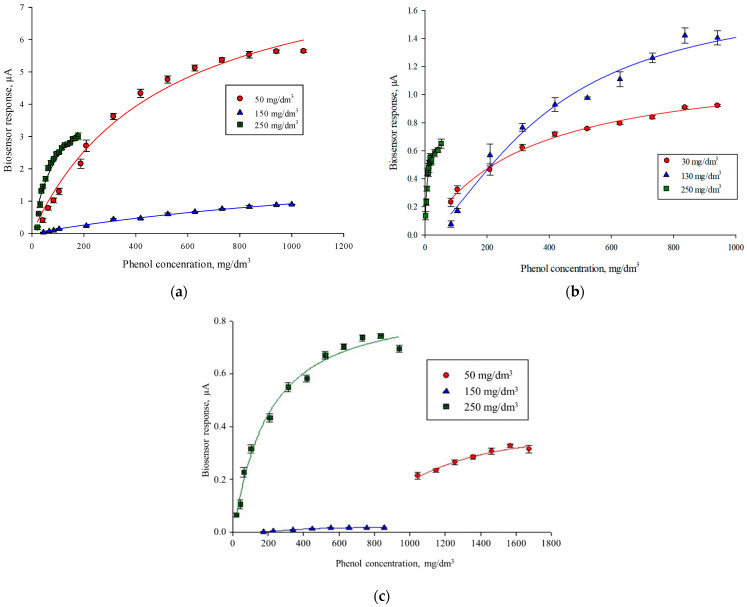
Calibration curves of the biosensor response on the concentration of phenol at different stages of adaptation of microorganisms: (**a**) *P. veronii* DSM 11331^T^; (**b**) *P. putida* BS394(pBS216); (**c**) *R. pyridinivorans* 5Ap.

**Figure 4 polymers-14-05366-f004:**
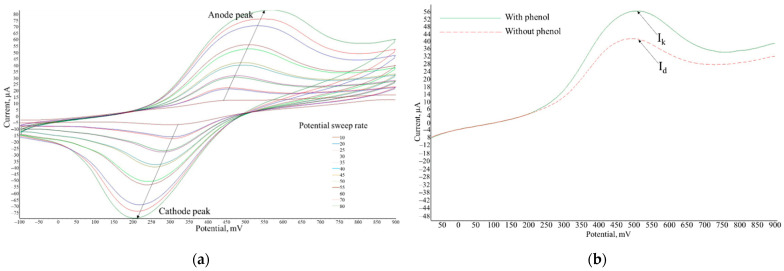
Typical current-voltage dependences for studying the electrochemical properties of the formed matrix: (**a**) voltammogram of the redox-active BSA-FC polymer with increasing scanning speed; (**b**) change in the strength of the anodic current in the presence and in the absence of the substrate for microorganisms *Pseudomonas putida* BS394(pBS216).

**Figure 5 polymers-14-05366-f005:**
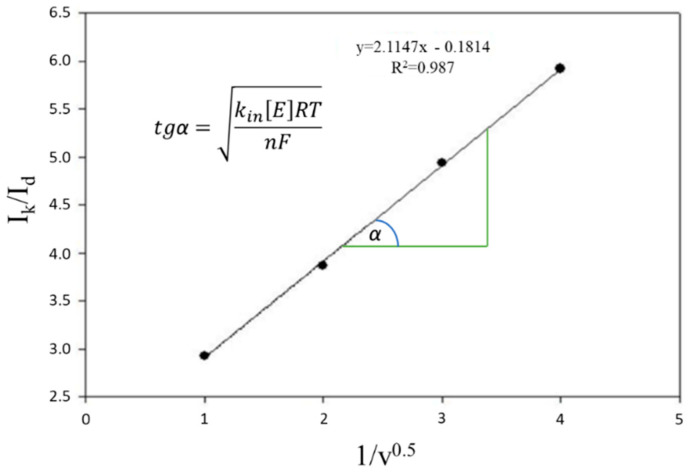
Determination of the mediator interaction constant with bacterial cells by the dependence of the limiting current on the sweep rate root *ν*^0.5^ in the *P. putida* BS394(pBS216)—BSA-ferrocene system.

**Figure 6 polymers-14-05366-f006:**
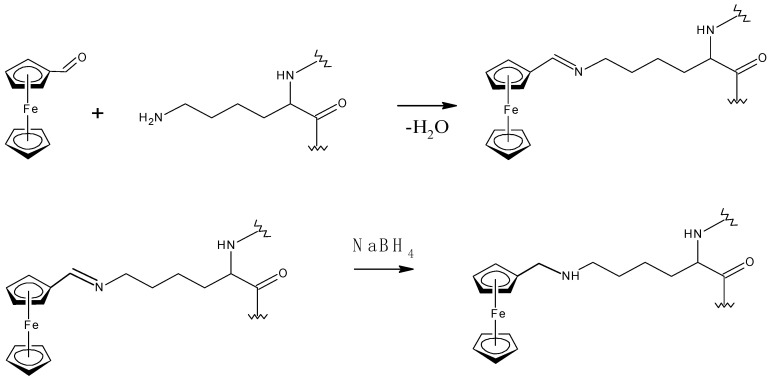
Scheme for the synthesis of a redox-active matrix based on BSA and ferrocene.

**Figure 7 polymers-14-05366-f007:**
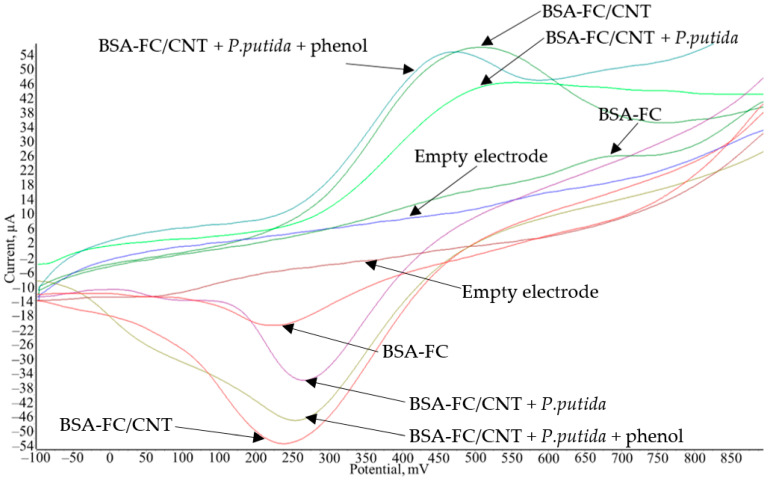
Voltammograms at different stages of electrode formation.

**Figure 8 polymers-14-05366-f008:**
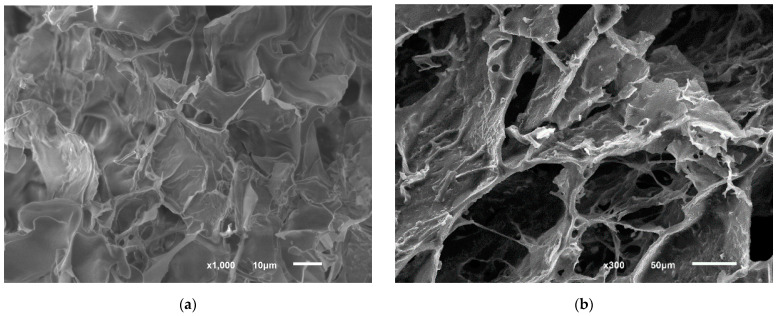
Scanning electron microscopy images for redox-active polymers: (**a**,**b**) BSA-FC; (**c**,**d**) BSA-FC/CNT; (**e**,**f**) BSA-FC/CNT with adapted *P. putida* BS394(pBS216).

**Figure 9 polymers-14-05366-f009:**
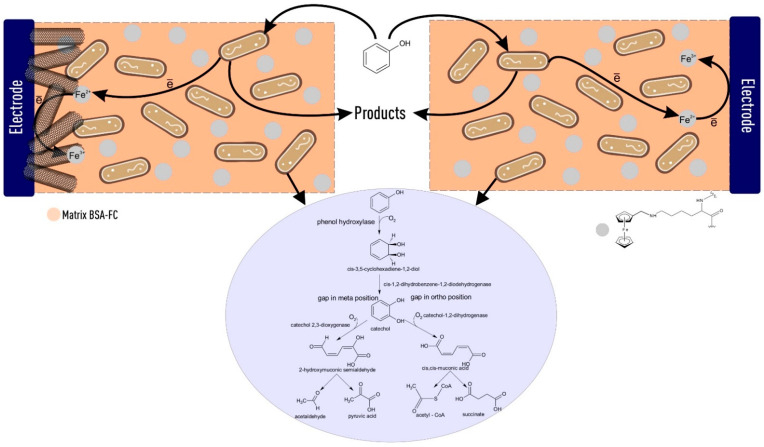
The process of electron transfer during the oxidation of phenol in the BSA-FC matrix and the BSA-FC/CNT matrix.

**Figure 10 polymers-14-05366-f010:**
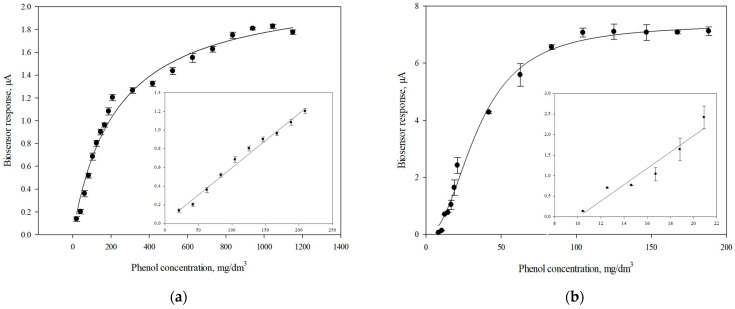
Calibrating curves of the biosensor response based on phenol-adapted bacteria *P. putida* BS394(pBS216) immobilized with template (**a**) BSA-ferrocenecarboxaldehyde using CNTs; (**b**) BSA-ferrocenecarboxaldehyde.

**Figure 11 polymers-14-05366-f011:**
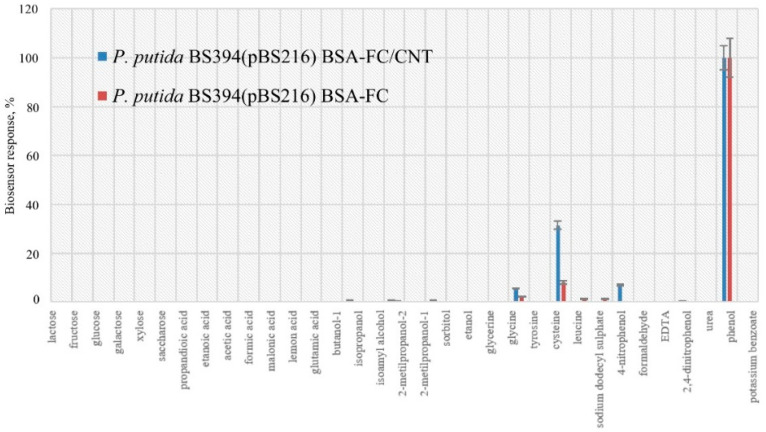
Substrate specificity of phenol-adapted microorganisms *P. putida* BS394(pBS216) immobilized in a gel based on ferrocenecarbaldehyde-modified BSA and BSA-FC/CNT using single-walled CNT.

**Table 1 polymers-14-05366-t001:** Main metrological and kinetic characteristics of receptor elements based on *P. veronii* DSM 11331^T^, *P. putida* BS394(pBS216) and *R. pyridinivorans* 5Ap at different stages of adaptation.

	Phenol/Glucose Concentration in the Growth Medium, mg/dm^3^	K_M_, mg/dm^3^	Lower Limit of Determined Concentrations, mg/dm^3^	Sensitivity Coefficient, 10^−3^ µA·dm^3^/mg	R_max_, µA
*P. veronii* DSM 11331^T^	50/200	500 ± 80	17.5	9.6 ± 0.7	1.24 ± 0.03
*P. veronii* DSM 11331^T^	150/100	1500 ± 200	30.2	1.10 ± 0.06	2.3 ± 0.2
*P. veronii* DSM 11331^T^	250/0	120 ± 10	35	7.2 ± 0.3	5.2 ± 0.3
*P. putida* BS394(pBS216)	30/220	330 ± 40	41.5	1.6 ± 0.3	8.8 ± 0.3
*P. putida* BS394(pBS216)	130/120	900 ± 300	36.5	2.9 ± 0.2	2.7 ± 0.5
*P. putida* BS394(pBS216)	250/0	6.7 ± 0.7	0.5	4.9 ± 0.3	0.73 ± 0.06
*R. pyridini-vorans* 5Ap	50/200	0.97 ± 0.03	1250	0.25 ± 0.04	0.36 ± 0.03
*R. pyridini-vorans* 5Ap	150/100	330 ± 50	216	0.45 ± 0.03	0.019 ± 0.002
*R. pyridine-vorans* 5Ap	250/0	219 ± 16	244	0.0031 ± 0.0002	0.92 ± 0.02

**Table 2 polymers-14-05366-t002:** Rate constants of interaction of electron transport mediators and a redox-active polymer with bacteria *P. putida* BS394(pBS216), which were found in this work.

Mediator	The Rate Constant of the Interaction of Mediators with Bacteria *P. putida* BS394(pBS216), dm^3^/(g·s)	Heterogeneous Rate Constant of Electron Transfer to a Graphite-Paste Electrode, Which Characterizes the Reaction Rate (3), cm/s
Neutral red	182.1 ± 0.4	0.017 ± 0.005 [35]
Thionin	33.8 ± 0.5	0.022 ± 0.005 [35]
Methylene blue	78.2 ± 0.3	0.025 ± 0.009 [35]
Potassium hexacyanoferrate (III)	162.01 ± 0.08	0.0067 ± 0.0009 [35]
Ferrocene	97.9 ± 0.2	0.4 ± 0.1 [35]
Ferrocenecarboxaldehyde	44.8 ± 0.2	0.03 ± 0.01 [35]
BSA-ferrocene	193.8 ± 0.7	0.45 ± 0.01 [30]
BSA-ferrocene/CNT	502.8 ± 0.6	0.55 ± 0.01 [30]
BSA-NR	3.9 ± 0.2	0.0119 ± 0.0006 [42]

**Table 3 polymers-14-05366-t003:** Analytical and metrological characteristics of the developed biosensors and their analogues.

Microorganism	Mediator System	K_M_, mg/dm^3^	Lower Limit of Determined Concentrations, mg/dm^3^	Sensitivity Coefficient, µA•dm^3^/mg	The Number of Oxidized Substrates	R_max_, µA
*P. putida* BS394(pBS216)	Ferrocene	7 ± 2	1.4	(0.05 ± 0.01) × 10^−3^	16	0.73 ± 0.06
*P. putida* BS394(pBS216)	BSA-ferrocene	35 ± 3	1 × 10^−3^	0.20 ± 0.04	5	7.4 ± 0.3
*P. putida* BS394(pBS216)	BSA-ferrocene/CNT	230 ± 30	1 × 10^−3^	(5.8 ± 0.2) × 10^−3^	6	2.17 ± 0.08
*Pseudomonas * sp. (GSN23) [30]	-	300	0.2	-	-	-
*Pseudomonas * sp. 74-III [43]	Duroquinone	0.094 ± 0.003	1 × 10^−4^	-	-	-
Consortium of *Bacillus* and *Pseudomonas* [44]	-	-	19.7	(0.95 ± 0.22) × 10^−3^	-	-

**Table 4 polymers-14-05366-t004:** Results of approbation of the created bioanalytical system.

Sample	Standard Method X ± ∆, mg/dm^3^	Biosensor Method X ± ∂, mg/dm^3^
River water	(3 ± 1) × 10^−3^	-
River water with a phenol concentration of 0.1 mg/dm^3^	0.10 ± 0.02	0.12 ± 0.02
River water with a phenol concentration of 5 mg/dm^3^	4.6 ± 0.5	4.5 ± 0.5
River water with a phenol concentration of 24 mg/dm^3^	23 ± 1	22 ± 2

## Data Availability

Not applicable.

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
