# Peer review of "Bioanalytical System for Determining the Phenol Index Based on *Pseudomonas putida* BS394(pBS216) Bacteria Immobilized in a Redox-Active Biocompatible Composite Polymer “Bovine Serum Albumin–Ferrocene–Carbon Nanotubes”"

_polymers, 2022, doi:10.3390/polym14245366_

Round 1

Reviewer 1 Report

Reviewer’s Comments:

The manuscript “Bioanalytical System for Determining the Phenol Index Based on Pseudomonas Putida BS394(pBS216) Bacteria Immobilized in a Redox-Active Biocompatible Composite Polymer "Bovine Serum Albumin - Ferrocene - Carbon Nanotubes” is a very interesting work. In this work, The possibility of using microorganisms Pseudomonas sp. 7р-81, Pseudomonas putida BS394(pBS216), Rhodococcus erythropolis s67, Rhodococcus pyridinivorans 5Ap, Rhodococcus erythropolis X5, Rhodococcus pyridinivorans F5, Pseudomonas veronii DSM 11331T as the basis of a biosensor for assessing the phenol index of aquatic environments was studied. The adaptation of microorganisms to phenol during growth was carried out to increase the selectivity of the analytical system. The most promising microorganisms for biosensor formation are the bacteria P. putida BS394(pBS216). Cells were immobilized in a redox ac-tive polymer based on bovine serum albumin with covalently bound ferrocene to increase sensitivity. While I believe this topic is of great interest to our readers, I think it needs minor revision before it is ready for publishing. So, I strongly recommend this manuscript for publication in this journal with minor revisions.

1. In abstract, the author should improve the quality of the abstract by adding more results.

2. Keywords: the synthesized system is missing in the keywords. So, modify the keywords.

3. In the introduction part, the author did not elaborate the scientific issues in this field and did not explain well that how their synthesized materials is suitable for these scientific issues.

4. Introduction part is not impressive. The references cited are very old. So, Improve it with some latest literature such as 10.1016/j.inoche.2022.109702, 10.3389/fmats.2022.875163

5. Selectivity of microorganisms in the adaptation process…, The author should provide reason about this statement “Mediators that are poorly soluble in water can be immobilized on the surface of the working electrode in order to modify the graphite paste.”.

6. The authors should explain regarding the recent literature why “This is probably due to the fact that adaptation is carried out in a medium containing not only phenol but also glucose as a carbon source”.

7. Material and method: Materials “write all the detail of chemicals in unique format rather than to write individual chemical such as Tin Chloride (SnCl4.5H2O). It should be written as “tin chloride (SnCl4.5H2O, 98%, Sigma)”. Write all the chemicals in this format.

8. In order check the stability of the synthesized materials, the authors should provide the SEM results after Bacteria Immobilized test.

9. Comparison of the present results with other similar findings in the literature should be discussed in more detail. This is necessary in order to place this work together with other work in the field and to give more credibility to the present results.

10. The conclusion part is very week. Improve by adding the results of your studies.

11. The authors should pay more attention to the English grammar, and the abbreviation of journal names in Ref.

Reviewer 2 Report

The paper is very well written and contributes a "Bovine Serum Albumin - Ferrocene - Carbon Nanotubes" scheme for Determining the Phenol Index, which enables accuracy and sensitivity. Furthermore, the proposed scheme outperforms the state of the arts and can be an appropriate and useful biosensor for the developed system as an analogue of the standard method for the determination of phenols in various samples, both natural and waste waters.

There are some problems, which must be solved before it is considered for publication. If the following problems are well-addressed, this reviewer believes that the essential contribution of this paper is important for Phenol Index Determining.

1. Reusability limitation

In-Page 14 lines 394-399, the authors demonstrated the stability and duration of operation of a biosensor. Long-term stability was measured over 15 days. That’s a very meaningful property of biosensors.

However, there is no test for reusability property. As we know, if the sensor could not reusable, it will significantly increase the cost of biosensors. So, the reviewer suggests adding more experiments for the reusability test.

2. SEM image limitation

In-Page 12 Figure 8, the authors said, “When using carbon nanotubes (Fig. 8b), they are uniformly distributed in the structure of the synthesized matrices.” However, there are not enough images to explain the results.

The authors only used one X10,000 SEM image, which is a small local image. The reviewer suggests adding more large area images, which show the whole structure.

3. Text editing

In-Page 5 Figure 1 and Page 6 Figure 2, the label size of the x-axis and y-axis are not uniform, it looks very strange for some are very big while some are small and not clear to read.

In-Page 6 Figure 2, the x-axis labels are very crowded and hard to read. The reviewer suggests keeping the same format as Figure 1. The labels with 90 degrees are easier to read.

In-Page 8 Table 1, the table title is not left justified with the table, it looks very strange. Follow your Page 10 Table 2, this format is good.

In-Page 14 Table 3, the same issue with table 1, not left justified.

In-Page 9 Figure 4, Page 12 Figures 8 and 9, Page 13 Figure 10, the figures have the same format issue as table 1, not left justified.

In-Page 13 Figure 11, the same issue with Figures 1 and 2.

In-Page 9 line 298, more space after Kin.
